# SSD: A Unified Framework for Self-Supervised Outlier Detection

**Vikash Sehwag**
Princeton University
`vvikash@princeton.edu`

**Mung Chiang**
Purdue University
`chiang@purdue.edu`

**Prateek Mittal**
Princeton University
`pmittal@princeton.edu`

## ABSTRACT

We ask the following question: what training information is required to design an effective outlier/out-of-distribution (OOD) detector, i.e., detecting samples that lie far away from the training distribution? Since unlabeled data is easily accessible for many applications, the most compelling approach is to develop detectors based on only unlabeled in-distribution data. However, we observe that most existing detectors based on unlabeled data perform poorly, often equivalent to a random prediction. In contrast, existing state-of-the-art OOD detectors achieve impressive performance but require access to fine-grained data labels for supervised training. We propose *SSD*, an outlier detector based on only unlabeled in-distribution data. We use self-supervised representation learning followed by a Mahalanobis distance based detection in the feature space. We demonstrate that *SSD* outperforms most existing detectors based on unlabeled data by a large margin. Additionally, *SSD* even achieves performance on par, and sometimes even better, with supervised training based detectors. Finally, we expand our detection framework with two key extensions. First, we formulate *few-shot OOD detection*, in which the detector has access to only one to five samples from each class of the targeted OOD dataset. Second, we extend our framework to incorporate training data labels, if available. We find that our novel detection framework based on *SSD* displays enhanced performance with these extensions, and achieves *state-of-the-art* performance[1].

## 1 INTRODUCTION

Deep neural networks are at the cornerstone of multiple safety-critical applications, ranging from autonomous driving (Ramanagopal et al., 2018) to biometric authentication (Masi et al., 2018; Günther et al., 2017). When trained on a particular data distribution, referred to as in-distribution data, deep neural networks are known to fail against test inputs that lie far away from the training distribution, commonly referred to as outliers or out-of-distribution (OOD) samples (Grubbs, 1969; Hendrycks & Gimpel, 2017). This vulnerability motivates the use of an outlier detector before feeding the input samples to the downstream neural network modules. However, a key question is to understand what training information is *crucial* for effective outlier detection? Will the detector require fine-grained annotation of training data labels or even access to a set of outliers in the training process?

Since neither data labels nor outliers are ubiquitous, the most compelling option is to design outlier detectors based on only *unlabeled* in-distribution data. However, we observe that most of the existing outlier detectors based on unlabeled data fail to scale up to complex data modalities, such as images. For example, autoencoder (AE) (Hawkins et al., 2002) based outlier detectors have achieved success in applications such as intrusion detection (Mirsky et al., 2018), and fraud detection (Schreyer et al., 2017). However, this approach achieves close to chance performance on image datasets. Similarly, density modeling based methods, such as PixelCNN++ (Salimans et al., 2017) and Glow (Kingma & Dhariwal, 2018) are known to assign even a higher likelihood to outliers in comparison to in-distribution data (Nalisnick et al., 2019).

In contrast, existing state-of-the-art OOD detectors achieve high success on image datasets but assume the availability of fine-grained labels for in-distribution samples (Hendrycks & Gimpel, 2017;

---

[1]Our code is publicly available at `https://github.com/inspire-group/SSD`

Bendale & Boult, 2016; Liang et al., 2018; Dhamija et al., 2018; Winkens et al., 2020). This is a strong assumption since labels, in-particular fine-grained labels, can be very costly to collect in some applications (Google AI Pricing, 2020), which further motivates the use of unlabeled data. The inability of supervised detectors to use unlabeled data and poor performance of existing unsupervised approaches naturally give rise to the following question.

> *Can we design an effective out-of-distribution (OOD) data detector with access to only unlabeled data from training distribution?*

A framework for outlier detection with unlabeled data[2] involves two key steps: 1) Learning a good feature representation with unsupervised training methods 2) Modeling features of in-distribution data without requiring class labels. For example, autoencoders attempt to learn the representation with a bottleneck layer, under the expectation that successful reconstruction requires learning a good set of representations. Though useful for tasks such as dimensionality reduction, we find that these representations are not good enough to sufficiently distinguish in-distribution data and outliers. We argue that if unsupervised training can develop a rich understanding of key semantics in in-distribution data then absence of such semantics in outliers can cause them to lie far away in the feature space, thus making it easy to detect them. Recently, self-supervised representation learning methods have made large progress, commonly measured by accuracy achieved on a downstream classification task (Chen et al., 2020; He et al., 2020; Oord et al., 2018; Misra & Maaten, 2020; Tian et al., 2020). We leverage these representations in our proposed cluster-conditioned framework based on the Mahalanobis distance (Mahalanobis, 1936). Our key result is that self-supervised representations are highly effective for the task of outlier detection in our self-supervised outlier detection (*SSD*) framework where they not only perform far better than most of the previous unsupervised representation learning methods but also perform on par, and sometimes even better, than supervised representations.

What if access to a fraction of OOD data or training data labels is available? How do we move past a detector based on unlabeled data and design a framework which can take advantage of such information? Though access to outliers during training is a strong assumption, it may be feasible to obtain a few prior instances of such outliers (Görnitz et al., 2013). We characterize this setting as *few-shot OOD detection*, where we assume access to very few, often one to five, samples from the targeted set of outliers. While earlier approaches (Liang et al., 2018; Lee et al., 2018b) mostly use such data to calibrate the detector, we find that access to just a few outliers can bring an additional boost in the performance of our detector. Crucial to this success is the reliable estimation of first and second order statistics of OOD data in the high dimensional feature space with just a few samples.

Finally, if class labels are available in the training phase, how can we incorporate them in the *SSD* framework for outlier detection? Recent works have proposed the addition of the supervised cross-entropy and self-supervised learning loss with a tunable parameter, which may require tuning for optimal parameter setting for each dataset (Hendrycks et al., 2019b; Winkens et al., 2020). We demonstrate that incorporating labels directly in the contrastive loss achieves 1) a tuning parameter-free detector, and 2) *state-of-the-art* performance.

## 1.1 KEY CONTRIBUTIONS

***SSD* for unlabeled data.** We propose *SSD*, an unsupervised framework for outlier detection based on unlabeled in-distribution data. We demonstrate that *SSD* outperforms most existing unsupervised outlier detectors by a large margin while also performing on par, and sometimes even better than supervised training based detection methods. We validate our observation across four different datasets: CIFAR-10, CIFAR-100, STL-10, and ImageNet.

**Extensions of *SSD*.** We provide two extensions of *SSD* to further improve its performance. First, we formulate *few-shot OOD detection* and propose detection methods which can achieve a significantly large gain in performance with access to only a few targeted OOD samples. Next, we extend *SSD*, without using any tuning parameter, to also incorporate in-distribution data labels and achieve *state-of-the-art* performance.

---

[2]We refer to OOD detection without using class labels of in-distribution data as unsupervised OOD detection.

## 2 RELATED WORK

**OOD detection with unsupervised detectors.** Interest in unsupervised outlier detection goes back to Grubbs (1969). We categorize these approaches in three groups 1) Reconstruction-error based detection using Auto-encoders (Hawkins et al., 2002; Mirsky et al., 2018; Schreyer et al., 2017) or Variational auto-encoders (Abati et al., 2019; An & Cho, 2015) 2) Classification based, such as Deep-SVDD (Ruff et al., 2018; El-Yaniv & Wiener, 2010; Geifman & El-Yaniv, 2017) and 3) Probabilistic detectors, such as density models like Glow and PixelCNN++ (Ren et al., 2019; Nalisnick et al., 2019; Salimans et al., 2017; Kingma & Dhariwal, 2018). We compare with detectors from each category and find that *SSD* outperforms them by a wide margin.

**OOD detection with supervised learning.** Supervised detectors have been most successful with complex input modalities, such as images and language (Chalapathy et al., 2018a; DeVries & Taylor, 2018; Dhamija et al., 2018; Jiang et al., 2018; Yoshihashi et al., 2018; Lee et al., 2018a). Most of these approaches model features of in-distribution data at output (Liang et al., 2018; Hendrycks & Gimpel, 2017; Dhamija et al., 2018) or in the feature space (Lee et al., 2018b; Winkens et al., 2020) for detection. We show that *SSD* can achieve performance on par with these supervised detectors, without using data labels. A subset of these detectors also leverages generic OOD data to boost performance (Hendrycks et al., 2019a; Mohseni et al., 2020).

**Access to OOD data at training time.** Some recent detectors also require OOD samples for hyperparameter tuning (Liang et al., 2018; Lee et al., 2018b; Zisselman & Tamar, 2020). We extend *SSD* to this setting but assume access to only a few OOD samples, referred to as *few-shot OOD detection*, which our framework can efficiently utilize to bring further gains in performance.

**In conjunction with supervised training.** Vyas et al. (2018) uses ensemble of leave-one-out classifier, Winkens et al. (2020) uses contrastive self-supervised training, and Hendrycks et al. (2019b) uses rotation based self-supervised loss, in conjunction with supervised cross-entropy loss to achieve state-of-the-art performance in OOD detection. Here we extend *SSD*, to incorporate data labels, when available, and achieve *better* performance than existing state-of-the-art.

**Anomaly detection**. In parallel to OOD detection, this research direction focuses on the detection of semantically related anomalies in applications such as intrusion detection, spam detection, disease detection, image classification, and video surveillance. We refer the interested reader to Pang et al. (2020) for a detailed review. While a large number of works focuses on developing methods particularly for single-class modeling in anomaly detection (Perera et al., 2019; Schlegl et al., 2017; Ruff et al., 2018; Chalapathy et al., 2018b; Golan & El-Yaniv, 2018; Wang et al., 2019), some recent work achieve success in both OOD detection and anomaly detection Tack et al. (2020); Bergman & Hoshen (2020). We provide a detailed comparison of our approach with previous work in both categories.

## 3 SSD: SELF-SUPERVISED OUTLIER/OUT-OF-DISTRIBUTION DETECTION

In this section, we first provide the necessary background on outlier/out-of-distribution (OOD) detection and then present the underlying formulation of our self-supervised detector (*SSD*) that relies on only unlabeled in-distribution data. Finally, we describe two extensions of *SSD* to (optionally) incorporate targeted OOD samples and in-distribution data labels (if available).

**Notation.** We represent the input space by $\mathcal{X}$ and corresponding label space as $\mathcal{Y}$. We assume in-distribution data is sampled from $\mathbb{P}^{in}_{\mathcal{X} \times \mathcal{Y}}$. In the absence of data labels, it is sampled from marginal distribution $\mathbb{P}^{in}_{\mathcal{X}}$. We sample out-of-distribution data from $\mathbb{P}^{ood}_{\mathcal{X}}$. We denote the feature extractor by $f : \mathcal{X} \rightarrow \mathcal{Z}$, where $\mathcal{Z} \subset \mathbb{R}^d$, a function which maps a sample from the input space to the $d$-dimensional feature space ($\mathcal{Z}$). The feature extractor is often parameterized by a deep neural network. In supervised learning, we obtain classification confidence for each class by $g \circ f : \mathcal{X} \rightarrow \mathbb{R}^c$. In most cases, $g$ is parameterized by a shallow neural network, generally a linear classifier.

**Problem Formulation: Outlier/Out-of-distribution (OOD) detection.** Given a collection of samples from $\mathbb{P}^{in}_{\mathcal{X}} \times \mathbb{P}^{ood}_{\mathcal{X}}$, the objective is to correctly identify the source distribution, i.e., $\mathbb{P}^{in}_{\mathcal{X}}$ or $\mathbb{P}^{ood}_{\mathcal{X}}$, for each sample. We use the term *supervised OOD detectors* for detectors which use in-distribution data labels, i.e., train the neural network ($g \circ f$) on $\mathbb{P}^{in}_{\mathcal{X} \times \mathcal{Y}}$ using supervised training techniques.

*Unsupervised OOD detectors* aim to solve the aforementioned OOD detection tasks, with access to only $\mathbb{P}_{\mathcal{X}}^{in}$. In this work, we focus on developing effective unsupervised OOD detectors.

**Background: Contrastive self-supervised representation learning.** Given unlabeled training data, it aims to train a feature extractor, by discriminating between individual instances from data, to learn a good set of representations. Using image transformations, it first creates two views of each image, commonly referred to as positives. Next, it optimizes to pull each instance close to its positive instances while pushing away from other images, commonly referred to as negatives. Assuming that $(x_i, x_j)$ are positive pairs for the $i$th image from a batch of $N$ images and $h(.)$ is a projection header, $\tau$ is the temperature, contrastive training minimizes the following loss, referred to as Normalized temperature-scaled cross-entropy (*NT-Xent*), over each batch.

$$\mathcal{L}_{batch} = \frac{1}{2N} \sum_{i=1}^{2N} -log \frac{e^{u_i^T u_j/\tau}}{\sum_{k=1}^{2N} \mathbb{1}(k \neq i) e^{u_i^T u_k/\tau}} \quad ; \qquad u_i = \frac{h\left(f(x_i)\right)}{\|h(f(x_i))\|_2} \tag{1}$$

### 3.1 UNSUPERVISED OUTLIER DETECTION WITH SSD

**Leveraging contrastive self-supervised training.** In the absence of data labels, *SSD* consists of two steps: 1) Training a feature extractor using unsupervised representation learning, 2) Developing an effective OOD detector based on hidden features which isn't conditioned on data labels.

We leverage contrastive self-supervised training for representation learning in our outlier detection framework, particularly due to its state-of-the-art performance (Chen et al., 2020; Tian et al., 2020). We will discuss the effect of different representation learning methods later in Section 4.2.

**Cluster-conditioned detection.** In absence of data labels, we develop a *cluster-conditioned* detection method in the feature space. We first partition the features for in-distribution training data in $m$ clusters. We represent features for each cluster as $\mathcal{Z}_m$. We use k-means clustering method, due to its effectiveness and low computation cost. Next, we model features in each cluster independently, and calculate the following *outlier score* $(s_x) = \min_m \mathcal{D}(x, \mathcal{Z}_m)$ for each test input $x$, where $\mathcal{D}(.,.)$ is a distance metric in the feature space. We discuss the choice of the number of clusters in Section 4.2.

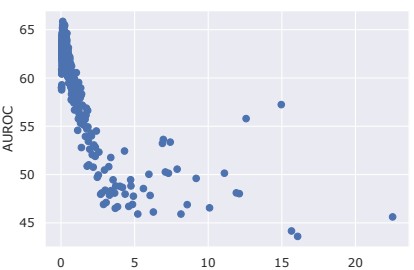

Figure 1: AUROC along individual principle eigenvector with CIFAR-10 as in-distribution and CIFAR-100 as OOD. Higher eigenvalues dominates euclidean distance, but are least helpful for outlier detection. Mahalnobis distance avoid this bias with appropriate scaling and performs much better.

**Choice of distance metric: Mahalanobis distance.** We use Mahalanobis distance to calculate the outlier score as follows:

$$s_x = \min_m (z_x - \mu_m)^T \Sigma_m^{-1} (z_x - \mu_m) \tag{2}$$

where $\mu_m$ and $\Sigma_m$ are the sample mean and sample covariance of features $(\mathcal{Z})$ of the in-distribution training data. We justify this choice with quantitative results in Figure 1. With eigendecomposition of sample covariance $(\Sigma_m = Q_m \Lambda_m Q_m^{-1})$, $s_x = \min_m \left(Q_m^T(z_x - \mu_m)\right)^T \Lambda_m^{-1} \left(Q_m^T(z_x - \mu_m)\right)$ which is equivalent to euclidean distance scaled with eigenvalues in the eigenspace. We discriminate between in-distribution (CIFAR-10) and OOD (CIFAR-100) data along each principal eigenvector (using AUROC, higher the better). With euclidean distance, i.e., in absence of scaling, components with higher eigenvalues have more weight but provide least discrimination. Scaling with eigenvalues removes the bias, making Mahalanobis distance effective for outlier detection in the feature space.

### 3.2 FEW-SHOT OOD DETECTION ($SSD_k$)

In this extension of the *SSD* framework, we consider the scenario where a few samples from the OOD dataset used at inference time are also available at the time of training. We focus on *one-shot* and *five-shot* detection, which refers to access to only one and fives samples, from each class of the targeted OOD dataset, respectively. Our hypothesis is that in-distribution samples and OOD samples

will be closer to other inputs from their respective distribution *in the feature space*, while lying further away from each other. We incorporate this hypothesis by using following formulation of outlier score.

$$s_x = (z_x - \mu_{in})^T \Sigma_{in}^{-1} (z_x - \mu_{in}) - (z_x - \mu_{ood})^T \Sigma_{ood}^{-1} (z_x - \mu_{ood}) \tag{3}$$

where $\mu_{in}, \Sigma_{in}$ and $\mu_{ood}, \Sigma_{ood}$ are the sample mean and sample covariance in the feature space for in-distribution and OOD data, respectively.

**Challenge.** The key challenge is to reliably estimate the statistics for OOD data, with access to only a few samples. Sample covariance is not an accurate estimator of covariance when the number of samples is less than the dimension of feature space (Stein, 1975), which is often in the order of thousands for deep neural networks.

**Shrunk covariance estimators and data augmentation.** We overcome this challenge by using following two techniques: 1) we use shrunk covariance estimators (Ledoit & Wolf, 2004), and 2) we amplify number of OOD samples using data augmentation. We use shrunk covariance estimators due to their ability to estimate covariance better than sample covariance, especially when the number of samples is even less than the feature dimension. To further improve the estimation we amplify the number of samples using data augmentation at the input stage. We use common image transformations, such as geometric and photometric changes to create multiple different images from a single source image from the OOD dataset. Thus given a set of $k$ OOD samples $\{u_1, u_2, \ldots, u_k\}$, we first create a set of $k \times n$ samples using data augmentation, $\mathcal{U} = \{u_1^1, \ldots, u_1^n, \ldots u_k^1, \ldots, u_k^n\}$. Using this set, we calculate the outlier score for a test sample in the following manner.

$$s_x = (z_x - \mu_{in})^T \Sigma_{in}^{-1} (z_x - \mu_{in}) - (z_x - \mu_U)^T S_U^{-1} (z_x - \mu_U) \tag{4}$$

where $\mu_U$ and $S_U$ are the sample mean and estimated covariance using shrunk covariance estimators for the set $\mathcal{U}$, respectively.

### 3.3    How to best use data labels ($SSD+$)

If fine-grained labels for in-distribution data are available, an immediate question is how to incorporate them in training to improve the success in detecting OOD samples.

**Conventional approach: Additive self-supervised and supervised training loss.** A common theme in recent works (Hendrycks et al. 2019b; Winkens et al. 2020) is to add self-supervised ($L_{ssl}$) and supervised ($L_{sup}$) training loss functions, i.e., $L_{training} = L_{sup} + \alpha L_{ssl}$, where the hyperparameter $\alpha$ is chosen for best performance on OOD detection. A common loss function for supervised training is cross-entropy.

**Our approach: Incorporating labels in contrastive self-supervised training.** As we show in Section 4.2, even without labels, contrasting between instances using self-supervised learning is highly successful for outlier detection. We argue for a similar instance-based contrastive training, where labels can also be incorporated to further improve the learned representations. To this end, we use the recently proposed supervised contrastive training loss function (Khosla et al., 2020), which uses labels for a more effective selection of positive and negative instances for each image. In particular, we minimize the following loss function.

$$\mathcal{L}_{batch} = \frac{1}{2N} \sum_{i=1}^{2N} -log \frac{\frac{1}{2N_{y_i}-1} \sum_{k=1}^{2N} \mathbb{1}(k \neq i) \mathbb{1}(y_k = y_i) e^{u_i^T u_k / \tau}}{\sum_{k=1}^{2N} \mathbb{1}(k \neq i) e^{u_i^T u_k / \tau}} \tag{5}$$

where $N_{y_i}$ refers to number of images with label $y_i$ in the batch. In comparison to contrastive NT-Xent loss (Equation 1), now we use images with identical labels in each batch as positives. We will show the superior performance of this approach compared to earlier approaches, and note that it is also a parameter-free approach which doesn't require additional OOD data to tune parameters. We further use the proposed cluster-conditioned framework with Mahalnobis distance, as we find it results in better performance than using data labels. We further summarize our framework in Algorithm 1.

---

**Algorithm 1:** Self-supervised outlier detection framework (*SSD*)

---

**Input** : $\mathcal{X}_{in}$, $\mathcal{X}_{test}$, feature extractor ($f$), projection head ($h$), Required True-positive rate ($T$),
    *Optional*: $\mathcal{X}_{ood}$, $\mathcal{Y}_{in}$ # $\mathcal{X}_{in} \in \mathbb{P}_X^{in}$, $\mathcal{X}_{ood} \in \mathbb{P}_X^{ood}$

**Output :** Is outlier or not? $\forall x \in \mathcal{X}_{test}$

**Function** *getFeatures($\mathcal{X}$): return* $\{f(x_i)/\|f(x_i)\|_2, \forall x_i \in \mathcal{X}\}$;

**Function** *SSDScore($\mathcal{Z}, \mu, \Sigma$): return* $\{(z-\mu)^T\Sigma^{-1}(z-\mu), \forall z \in \mathcal{Z}\}$;

**Function** *SSDkScore($\mathcal{Z}, \mu_{in}, \Sigma_{in}, \mu_{ood}, \Sigma_{ood}$):*
  | return $\{(z-\mu_{in})^T\Sigma_{in}^{-1}(z-\mu_{in}) - (z-\mu_{ood})^T\Sigma_{ood}^{-1}(z-\mu_{ood}), \forall z \in \mathcal{Z} \}$;

**end**

Parition $\mathcal{X}_{in}$ in training set ($\mathcal{X}_{train}$) and calibration set ($\mathcal{X}_{cal}$);

**if** $\mathcal{Y}_{in}$ *is not available* **then**

  | $\mathcal{L}_{batch} = \frac{1}{2N}\sum_{i=1}^{2N} -log\frac{e^{u_i^T u_j/\tau}}{\sum_{k=1}^{2N} 1(k \neq i)e^{u_i^T u_k/\tau}}$; $u_i = \frac{h(f(x_i))}{\|h(f(x_i))\|_2}$; # Train feature extractor

**else**

  | $\mathcal{L}_{batch} = \frac{1}{2N}\sum_{i=1}^{2N} -log\frac{\frac{1}{2N_{y_i}-1}\sum_{k=1}^{2N} 1(k \neq i)1(y_k=y_i)e^{u_i^T u_k/\tau}}{\sum_{k=1}^{2N} 1(k \neq i)e^{u_i^T u_k/\tau}}$;

**end**

Train feature extractor ($f$) by minimizing $\mathcal{L}_{batch}$ over $\mathcal{X}_{train}$;

$\mathcal{Z}_{train} = \texttt{getFeatures}(\mathcal{X}_{train})$, $\mathcal{Z}_{cal} = \texttt{getFeatures}(\mathcal{X}_{cal})$

$\mathcal{Z}_{test} = \texttt{getFeatures}(\mathcal{X}_{test})$, **if** $\mathcal{X}_{ood}$ *is available*: $\mathcal{Z}_{ood} = \texttt{getFeatures}(\mathcal{X}_{ood})$;

**if** $\mathcal{X}_{ood}$ *is not available* **then**

  | $s_{cal} = \texttt{SSDScore}(\mathcal{Z}_{cal}, \mu_{train}, \Sigma_{train})$; # Sample mean and convariance of $\mathcal{Z}_{train}$
  | $s_{test} = \texttt{SSDScore}(\mathcal{Z}_{test}, \mu_{train}, \Sigma_{train})$ # outlier score;

**else**

  | # Using convariance estimation techniques from Section 3.2 for $\mathcal{Z}_{ood}$
  | $s_{cal} = \texttt{SSDkScore}(\mathcal{Z}_{cal}, \mu_{train}, \Sigma_{train}, \mu_{ood}, \Sigma_{ood})$;
  | $s_{test} = \texttt{SSDkScore}(\mathcal{Z}_{test}, \mu_{train}, \Sigma_{train}, \mu_{ood}, \Sigma_{ood})$;

**end**

$x_i \in \mathcal{X}_{test}$ is an outlier if $s_{test}^i > (s_{cal}$ threshold at TPR = $T$);

---

# 4 EXPERIMENTAL RESULTS

## 4.1 COMMON SETUP ACROSS ALL EXPERIMENTS

We use recently proposed NT-Xent loss function from SimCLR (Chen et al., 2020) method for self-supervised training. We use the ResNet-50 network in all key experiments but also provide ablation with ResNet-18, ResNet-34, and ResNet-101 network architecture. We train each network, for both supervised and self-supervised training, with stochastic gradient descent for 500 epochs, 0.5 starting learning rate with cosine decay, and weight decay and batch size set to 1e-4 and 512, respectively. We set the temperature parameter to 0.5 in the NT-Xent loss. We evaluate our detector with three performance metrics, namely FPR (at TPR=95%), AUROC, and AUPR. For the supervised training baseline, we use identical training budget as *SSD* while also using the Mahalanobis distance based detection in the feature space. Due to space constraints, we present results with AUROC, which refers to area under the receiver operating characteristic curve, in the main paper and provide detailed results with other performance metrics in Appendix B.5. Our setup incorporates six image datasets along with additional synthetic datasets based on random noise. We report the average results over three independent runs in most experiments.

**Number of clusters.** We find the choice of the number of clusters dependent on which layer we extract the features from in the Residual neural networks. While for the first three blocks, we find an increase in AUROC with the number of clusters, the trend is reversed for the last block (Appendix B.2). Since the last block features achieve the highest detection performance, we model the in-distribution features as a single cluster in subsequent experiments.

## 4.2 PERFORMANCE OF SSD

**Comparison with unsupervised learning based detectors.** We present this comparison in Table 1. We find that *SSD* improves average AUROC by up to 55, compared to standard outlier detectors

Table 1: Comparison of *SSD* with different outlier detectors using only unlabeled training data.

| In-distribution (Out-of-distribution) | CIFAR-10 (SVHN) | CIFAR-10 (CIFAR-100) | CIFAR-100 (SVHN) | CIFAR-100 (CIFAR-10) | Average |
|---|---|---|---|---|---|
| Autoencoder (Hawkins et al., 2002) | 2.5 | 51.3 | 3.0 | 51.4 | 27.0 |
| VAE (Kingma & Welling, 2014) | 2.4 | 52.8 | 2.6 | 47.1 | 26.2 |
| PixelCNN++ (Salimans et al., 2017) | 15.8 | 52.4 | – | – | – |
| Deep-SVDD (Ruff et al., 2018) | 14.5 | 52.1 | 16.3 | 51.4 | 33.5 |
| Rotation-loss (Gidaris et al., 2018) | 97.9 | 81.2 | 94.4 | 50.1 | 80.9 |
| CSI (Tack et al., 2020) | **99.8** | 89.2 | – | – | – |
| *SSD* | 99.6 | **90.6** | **94.9** | **69.6** | **88.7** |
| $SSD_k$ ($k = 5$) | **99.7** | **93.1** | **99.1** | **78.2** | **92.5** |

based on Density modeling (PixelCNN++ (Salimans et al., 2017)), input reconstruction (Autoencoder (Hawkins et al., 2002), Variational Auto-encoder (Kingma & Welling, 2014)), and One-class classification (Deep-SVDD Ruff et al. (2018)). A common limitation of each of these three detectors is to find images from the SVHN dataset as more in-distribution when trained on CIFAR-10 or CIFAR-100 dataset. In contrast, *SSD* is able to successfully detect a large fraction of outliers from SVHN dataset. We also experiment with Rotation-loss Gidaris et al. (2018), a non-contrastive self-supervised training objective. We find that *SSD* with contrastive NT-Xent loss achieves 9.6% higher average AUROC compared to using Rotation-loss.

**Ablation studies.** We ablate along individual parameters in self-supervised training with CIFAR-10 as in-distribution data (Figure 2). While architecture doesn't have a very large effect on AUROC for most OOD dataset, we find that the number of training epochs and batch size plays a key role in detecting outliers from the CIFAR-100 dataset, which is hardest to detect among the four OOD datasets. We also find an increase in the size of training dataset helpful in the detection of all four OOD datasets.

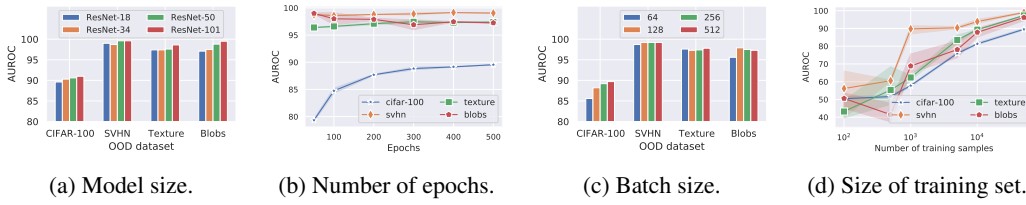

|  (a) Model size. | (b) Number of epochs. | (c) Batch size. | (d) Size of training set. |
|---|---|---|---|

Figure 2: Ablating across different training parameters in *SSD* under following setup: In-distribution dataset = CIFAR-10, OOD dataset = CIFAR-100, Training epochs = 500, Batch size = 512.

**Comparison with supervised representations.** We earlier asked the question *whether data labels are even necessary to learn representations crucial for OOD detection?* To answer it, we compare *SSD* with a supervised network, trained with an identical budget as *SSD* while also using Mahalanobis distance in the feature space, across sixteen different pairs of in-distribution and out-of-distribution datasets (Table 2). We observe that self-supervised representations even achieve better performance than supervised representations for 56% of the tasks in Table 2.

**Success in anomaly detection**. We now measure the performance of *SSD* in anomaly detection where we consider one of the CIFAR-10 classes as in-distribution and the rest of the classes as a source of anomalies. Similar to the earlier setup, we train a ResNet-50 network using self-supervised training with NT-Xent loss function. While the contrastive loss attempt to separate individual instances in the feature space, we find that adding an $\ell_2$ regularization in the feature space helps in improving performance. In particular, we add this regularization (with a scaling coefficient of 0.01) to bring individual instance features close to the mean of all feature vectors in the batch. Additionally, we reduce the temperature from 0.5 to 0.1 to reduce the separability of individual instances due to the contrastive loss. Overall, we find that our approach outperforms all previous works and achieves competitive performance with the concurrent work of Tack et al. (2020) (Table 3).

### 4.3 FEW-SHOT OOD DETECTION ($SSD_k$)

**Setup.** We focus on one-shot and five-shot OOD detection, i.e., set $k$ to 1 or 5 in Equation 4 and use Ledoit-Wolf (Ledoit & Wolf, 2004) estimator for covariance estimation. To avoid a bias on selected samples, we report average results over 25 random trials.

Table 2: Comparing performance of self-supervised (*SSD*) and supervised representations. We also provide results for few-shot OOD detection (*SSD$_k$*) for comparison with our baseline *SSD* detector.

| In-distribution | OOD | *SSD* | Supervised | *SSD$_k$* | | In-distribution | OOD | *SSD* | Supervised | *SSD$_k$* | |
| | | | | k=1 | k=5 | | | | | k=1 | k=5 |
|---|---|---|---|---|---|---|---|---|---|---|---|
| CIFAR-10 | CIFAR-100 | 90.6 | 90.6 | 91.7 | 93.0 | STL-10 | CIFAR-100 | **94.8** | 84.0 | 90.1 | 90.0 |
| | SVHN | 99.6 | 99.6 | 99.9 | 99.7 | | SVHN | **98.7** | 95.7 | 98.7 | 99.4 |
| | Texture | 97.6 | **97.8** | 98.9 | 99.4 | | Texture | **85.8** | 75.5 | 85.7 | 84.5 |
| | Blobs | 98.8 | **99.9** | 99.7 | 100.0 | | Blobs | **96.4** | 88.6 | 96.5 | 99.9 |
| | LSUN | **96.5** | 93.8 | 97.6 | 97.8 | | LSUN | **88.8** | 66.8 | 94.1 | 94.5 |
| | Places365 | 95.2 | 92.7 | 96.7 | 97.3 | | Places365 | **88.3** | 64.9 | 95.4 | 95.6 |
| CIFAR-100 | CIFAR-10 | **69.6** | 55.3 | 74.8 | 78.3 | ImageNet | SVHN | 99.1 | **99.4** | 99.7 | 100.0 |
| | SVHN | **94.9** | 94.5 | 99.5 | 99.1 | | Texture | 95.4 | 85.1 | 94.7 | 97.3 |
| | Texture | 82.9 | **98.8** | 96.8 | 94.2 | | Blobs | **99.5** | 98.4 | 100.0 | 100.0 |
| | Blobs | **98.1** | 57.3 | 98.1 | 100.0 | | Gaussian Noise | 100.0 | 100.0 | 100.0 | 100.0 |
| | LSUN | **79.5** | 69.4 | 92.3 | 93.4 | | ImageNet-O | 45.2 | **75.5** | 89.4 | 93.3 |
| | Places365 | **79.6** | 62.6 | 90.7 | 92.7 | | | | | | |

Table 3: Comparison of *SSD* with other detectors for anomaly detection task on CIFAR-10 dataset.

| | Airplane | Automobile | Bird | Cat | Deer | Dog | Frog | Horse | Ship | Truck | Average |
|---|---|---|---|---|---|---|---|---|---|---|---|
| Randomly Initialized network | 77.4 | 44.1 | 62.4 | 44.1 | 62.1 | 49.6 | 59.8 | 48.0 | 73.8 | 53.7 | 57.5 |
| VAE (Kingma & Welling, 2014) | 70.0 | 38.6 | 67.9 | 53.5 | 74.8 | 52.3 | 68.7 | 49.3 | 69.6 | 38.6 | 58.3 |
| OCSVM (Schölkopf et al., 2001) | 63.0 | 44.0 | 64.9 | 48.7 | 73.5 | 50.0 | 72.5 | 53.3 | 64.9 | 50.8 | 58.5 |
| AnoGAN (Schlegl et al., 2017) | 67.1 | 54.7 | 52.9 | 54.5 | 65.1 | 60.3 | 58.5 | 62.5 | 75.8 | 66.5 | 61.8 |
| PixelCNN (Van den Oord et al., 2016) | 53.1 | 99.5 | 47.6 | 51.7 | 73.9 | 54.2 | 59.2 | 78.9 | 34.0 | 66.2 | 61.8 |
| DSVDD (Ruff et al., 2018) | 61.7 | 65.9 | 50.8 | 59.1 | 60.9 | 65.7 | 67.7 | 67.3 | 75.9 | 73.1 | 64.8 |
| OCGAN (Perera et al., 2019) | 75.7 | 53.1 | 64.0 | 62.0 | 72.3 | 62.0 | 72.3 | 57.5 | 82.0 | 55.4 | 65.6 |
| RCAE (Chalapathy et al., 2018b) | 72.0 | 63.1 | 71.7 | 60.6 | 72.8 | 64.0 | 64.9 | 63.6 | 74.7 | 74.5 | 68.2 |
| DROCC (Goyal et al., 2020) | 81.7 | 76.7 | 66.7 | 67.1 | 73.6 | 74.4 | 74.4 | 71.4 | 80.0 | 76.2 | 74.2 |
| Deep-SAD (Ruff et al., 2019) | – | – | – | – | – | – | – | – | – | – | 77.9 |
| E3Outlier (Wang et al., 2019) | 79.4 | 95.3 | 75.4 | 73.9 | 84.1 | 87.9 | 85.0 | 93.4 | 92.3 | 89.7 | 85.6 |
| GT (Golan & El-Yaniv, 2018) | 74.7 | 95.7 | 78.1 | 72.4 | 87.8 | 87.8 | 83.4 | 95.5 | 93.3 | 91.3 | 86.0 |
| InvAE (Huang et al., 2019) | 78.5 | 89.8 | 86.1 | 77.4 | 90.5 | 84.5 | 89.2 | 92.9 | 92.0 | 85.5 | 86.6 |
| GOAD (Bergman & Hoshen, 2020) | 77.2 | 96.7 | 83.3 | 77.7 | 87.8 | 87.8 | 90.0 | 96.1 | 93.8 | 92.0 | 88.2 |
| CSI (Tack et al., 2020) | 89.9 | 99.9 | 93.1 | 86.4 | 93.9 | 93.2 | 95.1 | 98.7 | 97.9 | 95.5 | 94.3 |
| *SSD* | 82.7 | 98.5 | 84.2 | 84.5 | 84.8 | 90.9 | 91.7 | 95.2 | 92.9 | 94.4 | 90.0 |

**Results.** Compared to the baseline *SSD* detector, one-shot and five-shot settings improve the average AUROC, across all OOD datasets, by 1.6 and 2.1, respectively (Table 1, 2). In particular, we observe large gains with CIFAR-100 as in-distribution and CIFAR-10 as OOD where five-shot detection improves the AUROC from 69.6 to 78.3. We find the use of shrunk covariance estimator most critical in the success of our approach. Use of shrunk covariance estimation itself improves the AUROC from 69.6 to 77.1. Then data augmentation further improves it 78.3 for the five-shot detection. With an increasing number of transformed copies of each sample, we also observe improvement in AUROC, though it later plateaus close to ten copies (Appendix B.3).

**What if additional OOD images are available** Note that some earlier works, such as Liang et al. (2018), assume that 1000 OOD inputs are available for tuning the detector. We find that with access to this large amount of OOD samples, *SSD$_k$* can improve the state-of-the-art by an even larger margin. For example, with CIFAR-100 as in-distribution and CIFAR-10 as out-of-distribution, it achieves 89.4 AUROC, which is 14.2% higher than the current state-of-the-art (Winkens et al., 2020).

## 4.4 Success when using data labels (*SSD+*)

Now we integrate labels of training data in our framework and compare it with the existing state-of-the-art detectors. We report our results in Table 4. Our approach improves the average AUROC by 0.8 over the previous state-of-the-art detector. Our approach also achieves equal or better performance than previous state-of-the-art across individual pairs of in and out-distribution dataset. For example, using labels in our framework improves the AUROC of Mahalanobis detector from 55.5 to 72.1 for CIFAR-100 as in-distribution and CIFAR-10 as the OOD dataset. Using the simple softmax probabilities, training a two-layer MLP on learned representations further improves the AUROC to 78.3. Combining *SSD+* with a five-shot OOD detection method further brings a gain of 1.4 in the average AUROC.

## 5 Discussion and Conclusion

**On tuning hyperparameters in *SSD*.** In our framework, we either explicitly avoid the use of additional tuning-parameters (such as when combining self-supervised and supervised loss functions

Table 4: Comparison of *SSD+*, i.e., incorporating labels in the *SSD* detector, with state-of-the-art detectors based on supervised training.

| In-distribution (Out-of-distribution) | CIFAR-10 (CIFAR-100) | CIFAR-10 (SVHN) | CIFAR-100 (CIFAR-10) | CIFAR-100 (SVHN) | Average |
|---|---|---|---|---|---|
| Softmax-probs (Hendrycks & Gimpel, 2017) | 89.8 | 95.9 | 78.0 | 78.9 | 85.6 |
| ODIN(Liang et al., 2018)† | 89.6 | 96.4 | 77.9 | 60.9 | 81.2 |
| Mahalnobis (Lee et al., 2018b)† | 90.5 | 99.4 | 55.3 | 94.5 | 84.8 |
| Residual Flows (Zisselman & Tamar, 2020)† | 89.4 | 99.1 | 77.1 | **97.5** | 90.7 |
| Gram Matrix (Sastry & Oore, 2019) | 79.0 | 99.5 | 67.9 | 96.0 | 85.6 |
| Outlier exposure (Hendrycks et al., 2019a) | 93.3 | 98.4 | 75.7 | 86.9 | 88.6 |
| Rotation-loss + Supervised (Hendrycks et al., 2019b) | 90.9 | 98.9 | – | – | – |
| Contrastive + Supervised (Winkens et al., 2020)* | 92.9 | 99.5 | **78.3** | 95.6 | 91.6 |
| CSI (Tack et al., 2020) | 92.2 | 97.9 | – | – | – |
| *SSD+* | **93.4** | **99.9** | **78.3** | **98.2** | **92.4** |
| $SSD_k+$ ($k = 5$) | **94.1** | **99.6** | **84.1** | **97.4** | **93.8** |

\* Uses 4× wider ResNet-50 network, † Requires additional out-of-distribution data for tuning.

in *SSD+*) or refrain from tuning the existing set of parameters for each OOD dataset. For example, we use a standard set of parameters for self-supervised training and model the learned features with a single-cluster.

**Why contrastive self-supervised learning is effective in the *SSD* framework?** We focus on the NT-Xent loss function, which is parameterized by a temperature variable ($\tau$). Its objective is to pull positive instances, i.e., different transformations of an image, together while pushing away from other instances. Earlier works have shown that such contrastive training forces the network to learn a good set of feature representations. However, a smaller value of temperature quickly saturates the loss, discouraging it to further improve the feature representations. We find that

Table 5: Test Accuracy and AUROC with different temperature values in *NT-Xent* (Equation 1) loss. Using CIFAR-10 as in-distribution and CIFAR-100 as OOD dataset with ResNet18 network.

| Temperature | 0.001 | 0.01 | 0.1 | 0.5 |
|---|---|---|---|---|
| Test -Accuracy | 70.8 | 76.7 | 86.9 | 90.5 |
| AUROC | 66.7 | 71.6 | 85.5 | 89.5 |

the performance of *SSD* also degrades with lower temperature, suggesting the necessity of learning a good set of feature representation for effective outlier detection (Table 5).

**How discriminative ability of feature representations evolves over the course of training.** We analyze this effect in Figure 3 where we compare both *SSD* and supervised training based detector over the course of training. While discriminative ability of self-supervised training in *SSD* is lower at the start, it quickly catches up with supervised representations after half of the training epochs.

**Performance of *SSD* improves with the amount of available unlabeled data.** A compelling advantage of unsupervised learning is to learn from unlabeled data, which can be easily collected. As presented in Figure 2, we find that performance of *SSD* increases with the size of training dataset. We conduct another experiment with the STL-10 dataset, where in addition to the 5k training images, we also use additional 10k images from the unlabeled set. This improves the AUROC from 94.7 to 99.4 for CIFAR-100 as the OOD dataset, further demonstrating the success of *SSD* in leveraging unlabeled data (Appendix B.4). In conclusion, our framework provides an effective & flexible approach for outlier detection using unlabeled data.

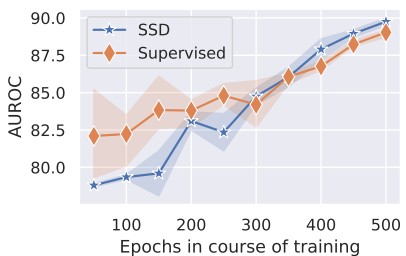

Figure 3: AUROC over the course of training with CIFAR-10 as in-distribution and CIFAR-100 as OOD set.

**Acknowledgments** We would like to thank Chong Xiang, Liwei Song, and Arjun Nitin Bhagoji for their helpful feedback on the paper. This work was supported in part by the National Science Foundation under grants CNS-1553437 and CNS-1704105, by a Qualcomm Innovation Fellowship, by the Army Research Office Young Investigator Prize, by Army Research Laboratory (ARL) Army Artificial Intelligence Institute (A2I2), by Office of Naval Research (ONR) Young Investigator Award, by Facebook Systems for ML award, by Schmidt DataX Fund, and by Princeton E-ffiliates Partnership.

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

# A ADDITIONAL DETAILS ON EXPERIMENTAL SETUP

## A.1 TRAINING AND EVALUATION SETUP FOR DEEP NEURAL NETWORKS.

We use ResNet-50 architecture for all our major experiments and ResNet-18 for ablation studies. We also provide results with ResNet-34 and ResNet-101 architecture. We use a two-layer fully connected network as the projection header ($h(.)$). To contrast with a large number of negatives, NT-Xent loss requires a much larger batch size compared to the supervised cross-entropy loss function. We train it using a batch size of 512. When evaluating self-supervised models, even when we incorporate labels in *SSD*, we achieve the best performance when modeling in-distribution features with only a single cluster. However, for supervised training, which refers to the supervised baseline in the paper, we find that increasing the number of clusters helps. For it, we report the best of the results obtained from cluster indexes or using true labels of the data. For each dataset, we use the test set partition, if it exists, as the OOD dataset. For consistent comparison, we re-implement Softmax-probabilities (Hendrycks & Gimpel, 2017), ODIN (Liang et al., 2018), and Mahalanobis detector (Lee et al., 2018b) and evaluate their performance on the identical network, trained with supervised training for 500 epochs. We set the perturbation budget to 0.0014 and temperature to 1000 for ODIN, since these are the most successful set of parameters reported in the original paper (Liang et al., 2018). We primarily focus on one-shot and five-shot OOD detection, i.e., set $k$ to one or five. It implies access to one and five images, respectively, from each class of the targeted OOD dataset. We create ten randomly transformed samples from each available OOD image in the $SSD_k$ detector. With very small $k$, such as one, we find that increasing number of transformations may degrade performance in some cases. In this case we simply resort to using only one transformation per sample. To avoid any

hyperparameter selection, we set the image augmentation pipeline to be the same as the one used in training. Finally, we use the Ledoit-Wolf method (Ledoit & Wolf, 2004) to estimate the covariance of the OOD samples.

## A.2 PERFORMANCE METRICS FOR OUTLIER DETECTORS

We use the following three performance metrics to evaluate the performance of outlier detectors.

- **FPR at TPR=95%.** It refers to the false positive rate (= FP / (FP+TN)), when true positive rate (= TP / (TP + FN)) is equal to 95%. Effectively, its goal is to measure what fraction of outliers go undetected when it is desirable to have a true positive rate of 95%.

- **AUROC.** It refers to the area under the receiver operating characteristic curve. We measure it by calculating the area under the curve when we plot TPR against FPR.

- **AUPR.** It refers to the area under the precision-recall curve, where precision = TP / (TP+FP) and recall = TP / (TP+FN). Similar to AUROC, AUPR is also a threshold independent metric.

## A.3 DATASETS USED IN THIS WORK

We use the following datasets in this work. Whenever there is a mismatch between the resolution of images in in-distribution and out-of-distribution (OOD) data, we appropriately scale the OOD images with bilinear scaling. When there is an overlap between the classes of the in-distribution and OOD dataset, we remove the common classes from the OOD dataset.

- **CIFAR-10 (Krizhevsky et al., 2009)**. It consists of 50,000 training images and 10,000 test images from 10 different classes. Each image size is 32×32 pixels.

- **CIFAR-100 (Krizhevsky et al., 2009)**. CIFAR-100 also has 50,000 training images and 10,000 test images. However, it has 100 classes which are further organized in 20 sub-classes. Note that its classes aren't identical to the CIFAR-10 dataset, with a slight exception with class *truck* in CIFAR-10 and *pickup truck* in CIFAR-100. However, their classes share multiple similar semantics, making it hard to catch outliers from the other dataset.

- **SVHN (Netzer et al., 2011)**. SVHN is a real-world street-view housing number dataset. It has 73,257 digits available for training, and 26,032 digits for testing. Similar to the CIFAR-10/100 dataset, the size of its images is also 32×32 pixels.

- **STL-10 (Coates et al., 2011)**. STL-10 has identical classes as the CIFAR-10 dataset but focuses on the unsupervised learning. It has 5,000 training images, 8,000 test images, and a set of 100,000 unlabeled images. Unlike the previous three datasets, the size of its images is 96×96 pixels.

- **DTD (Cimpoi et al., 2014)**. Describable Textures Dataset (DTD) is a collection of textural images in the wild. It includes a total of 5,640 images, split equally between 47 categories where the size of images range between 300×300 and 640×640 pixels.

- **ImageNet[3] (Deng et al., 2009)**. ImageNet is a large scale dataset of 1,000 categories with 1.2 Million training images and 50,000 validation images. It has high diversity in both inter- and intra-class images and is known to have strong generalization properties to other datasets.

- **Blobs.** Similar to Hendrycks et al. (2019a), we algorithmically generate these amorphous shapes with definite edges.

- **Gaussian Noise.** We generate images with Gaussian noise using a mean of 0.5 and a standard deviation of 0.25. We clip the pixel value to the valid pixel range of [0, 1].

- **Uniform Noise.** It refers to images where each pixel value is uniformly sampled from the [0, 1] range.

---

[3]We refer to the commonly used ILSVRC 2012 release of ImageNet dataset.

## B   ADDITIONAL EXPERIMENTAL RESULTS

### B.1   LIMITATIONS OF OUTLIER DETECTORS BASED ON SUPERVISED TRAINING

Existing supervised training based detector assumes that fine-grained data labels are available for the training data. What happens to the performance of current detectors if we relax this assumption by assuming that only coarse labels are present. We simulate this setup by combining consecutive classes from the CIFAR-10 dataset into two groups, referred to as CIFAR-2, or five groups referred to as CIFAR-5. We use CIFAR-100 as the out-of-distribution dataset. We find that the performance of existing detectors degrades significantly when only coarse labels are present (Figure 4). In contrast, *SSD* operates on unlabeled data thus doesn't suffer from similar performance degradation.

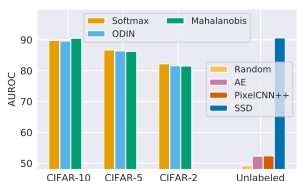

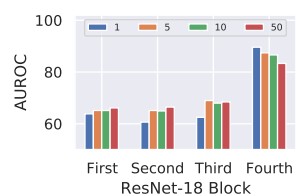

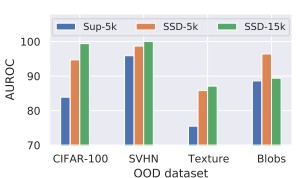

Figure 4: Existing supervised detector requires fine-grained labels. In contrast, *SSD* can achieve similar performance with only unlabeled data.

Figure 5: Relationship of AU-ROC with clusters depends on which block we use as the feature extractor.

Figure 6: Using extra unlabeled training data can help to further improve the performance of *SSD*.

### B.2   ON CHOICE OF NUMBER OF CLUSTERS

We find that the choice of optimal number of clusters is dependent on which layer we use as the feature extractor in a Residual Neural network. We demonstrate this trend in Figure 5, with CIFAR-10 as in-distribution dataset and CIFAR-100 as out-of-distribution dataset. We extract features from the last layer of each block in the residual network and measure the *SSD* performance with them. While for the first three blocks, we find an increase in AUROC with number of clusters, the trend is reversed for the last block (Figure 5). Since last block features achieve highest detection performance, we model in-distribution features using a single cluster.

### B.3   ABLATION STUDY FOR FEW-SHOT OOD DETECTION

For few shot OOD detection, we ablate along the number of transformations used for each sample. We choose CIFAR-100 as in-distribution and CIFAR-10 as OOD dataset with $SSD_k$, set $k$ to five, and choose ResNet-18 network architecture. When increasing number of transformations from 1, 5, 10, 20, 50 the AUROC of detector is 74.3, 75.7, 76.1, 76.3, 76.7. To achieve a balance between the performance and computational cost, we use ten transformations for each sample in our final experiments.

### B.4   PERFORMANCE OF SSD IMPROVES WITH AMOUNT OF UNLABELED DATA

With easy access to unlabeled data, it is compelling to develop detectors that can benefit from the increasing amount of such data. We earlier demonstrated this ability of *SSD* for the CIFAR-10 dataset in Figure 2. Now we present similar results with the STL-10 dataset. We first train a self-supervised network, and an equivalent supervised network with 5,000 training images from the STL-10 dataset. We refer to these networks by SSD-5k and Sup-5k, respectively. Next, we include additional 10,000 images from the available 100k unlabeled images in the dataset. As we show in Figure 6, *SSD* is able to achieve large gains in performance with access to the additional unlabeled training data.

### B.5   RESULTS WITH DIFFERENT PERFORMANCE METRICS

We provide our detailed experimental results for each component in the *SSD* framework with three different performance metrics in Table 6, 7,.

Table 6: Experimental results of *SSD* detector with multiple metrics for ImageNet dataset.

| In-distribution | OOD | FPR (TPR = 95%) ↓ | | | | AUROC ↑ | | | | AUPR ↑ | | | |
| | | *SSD* | Supervised | $SSD_k$ | | *SSD* | Superivsed | $SSD_k$ | | *SSD* | Supervised | $SSD_k$ | |
| | | | | k=1 | k=5 | | | k=1 | k=5 | | | k=1 | k=5 |
| ImageNet | SVHN | 1.3 | 0.6 | 0.0 | 0.0 | 99.4 | 99.1 | 100.0 | 100.0 | 98.4 | 96.6 | 100.0 | 100.0 |
| | Texture | 57.2 | 23.2 | 20.1 | 11.4 | 85.4 | 95.4 | 95.4 | 97.4 | 41.7 | 75.8 | 78.6 | 84.2 |
| | Blobs | 0.0 | 0.0 | 0.0 | 0.0 | 98.4 | 99.5 | 100.0 | 100.0 | 81.1 | 91.6 | 100.0 | 100.0 |
| | Gaussian noise | 0.0 | 0.0 | 0.0 | 0.0 | 100.0 | 100.0 | 100.0 | 100.0 | 100.0 | 100.0 | 100.0 | 100.0 |
| | Uniform noise | 0.0 | 0.0 | 0.0 | 0.0 | 100.0 | 100.0 | 100.0 | 100.0 | 100.0 | 100.0 | 100.0 | 100.0 |

Table 7: Experimental results of *SSD* detector with multiple metrics over CIFAR-10, CIFAR-100, and STL-10 dataset.

| In-distribution | OOD | FPR (TPR = 95%) ↓ | | | | | | | AUROC ↑ | | | | | | | AUPR ↑ | | | | | | |
|---|---|---|---|---|---|---|---|---|---|---|---|---|---|---|---|---|---|---|---|---|---|---|
| | | SSD | Supervised | $SSD_k$ k=1 | $SSD_k$ k=5 | SSD+ | $SSD_k$+ k=1 | $SSD_k$+ k=5 | SSD | Superivsed | $SSD_k$ k=1 | $SSD_k$ k=5 | SSD+ | $SSD_k$+ k=1 | $SSD_k$+ k=5 | SSD | Supervised | $SSD_k$ k=1 | $SSD_k$ k=5 | SSD+ | $SSD_k$+ k=1 | $SSD_k$+ k=5 |
| CIFAR-10 | CIFAR-100 | 50.7 | 47.4 | 44.7 | 39.4 | 38.5 | 36.3 | 34.6 | 90.6 | 90.6 | 91.7 | 93.0 | 93.4 | 93.4 | 94.0 | 89.2 | 89.5 | 90.5 | 91.9 | 92.3 | 92.5 | 92.9 |
| | SVHN | 2.0 | 1.6 | 0.2 | 1.0 | 0.2 | 0.5 | 1.9 | 99.6 | 99.6 | 99.9 | 99.7 | 99.9 | 99.9 | 99.6 | 99.8 | 99.8 | 100.0 | 100.0 | 99.9 | 100.0 | 99.8 |
| | Texture | 14.6 | 12.4 | 5.1 | 2.7 | 7.7 | 6.4 | 3.6 | 97.6 | 97.8 | 98.9 | 99.4 | 98.5 | 98.6 | 99.2 | 95.6 | 96.7 | 98.4 | 99.0 | 97.3 | 98.1 | 98.9 |
| | Blobs | 4.3 | 0.0 | 4.4 | 0.0 | 0.0 | 0.0 | 0.0 | 98.8 | 99.9 | 99.7 | 100.0 | 100.0 | 100.0 | 100.0 | 98.4 | 99.9 | 99.4 | 100.0 | 100.0 | 100.0 | 100.0 |
| CIFAR-100 | CIFAR-10 | 89.4 | 96.4 | 85.3 | 69.4 | 89.5 | 72.1 | 65.2 | 69.6 | 55.3 | 74.8 | 78.3 | 71.0 | 78.2 | 84.0 | 64.5 | 51.8 | 69.3 | 77.2 | 65.3 | 76.2 | 81.7 |
| | SVHN | 20.9 | 28.4 | 2.7 | 4.5 | 7.9 | 11.3 | 11.9 | 94.9 | 94.5 | 99.5 | 99.1 | 98.2 | 97.3 | 97.4 | 98.1 | 97.5 | 99.8 | 99.6 | 99.3 | 99.1 | 99.0 |
| | Texture | 65.8 | 2.3 | 16.4 | 25.1 | 68.1 | 24.3 | 23.8 | 82.9 | 98.8 | 96.8 | 94.2 | 81.2 | 93.8 | 94.5 | 72.9 | 97.9 | 95.0 | 91.9 | 70.7 | 91.6 | 92.2 |
| | Blobs | 1.2 | 95.6 | 1.1 | 0.0 | 3.6 | 0.0 | 0.0 | 98.1 | 57.3 | 98.1 | 100.0 | 98.8 | 100.0 | 100.0 | 97.8 | 47.6 | 97.8 | 100.0 | 98.2 | 100.0 | 100.0 |
| STL-10 | CIFAR-100 | 29.9 | 73.2 | 32.9 | 32.3 | 40.0 | 8.6 | 14.1 | 94.8 | 84.0 | 90.1 | 90.0 | 92.4 | 98.4 | 97.8 | 95.1 | 82.5 | 92.6 | 92.3 | 93.3 | 98.7 | 98.0 |
| | SVHN | 6.6 | 26.2 | 5.8 | 2.4 | 18.6 | 0.4 | 0.3 | 98.7 | 95.7 | 98.7 | 99.4 | 96.9 | 99.8 | 99.9 | 99.5 | 97.6 | 99.5 | 99.7 | 98.9 | 100.0 | 100.0 |
| | Texture | 53.0 | 69.4 | 50.0 | 39.5 | 51.8 | 46.2 | 43.8 | 85.8 | 75.5 | 85.7 | 84.5 | 85.8 | 90.4 | 91.0 | 82.6 | 69.3 | 84.1 | 84.3 | 83.7 | 87.1 | 87.7 |
| | Blobs | 16.3 | 88.9 | 14.6 | 0.0 | 67.8 | 0.1 | 0.0 | 96.4 | 88.6 | 96.5 | 99.9 | 92.9 | 99.7 | 99.8 | 92.1 | 81.3 | 92.2 | 99.9 | 86.7 | 99.5 | 99.7 |

