# OpenReview forum: "SSD: A Unified Framework for Self-Supervised Outlier Detection"
_ICLR.cc/2021/Conference — ICLR 2021 Poster_

### Official Review · AnonReviewer3 · 2020-10-27
**Good paper**

**Rating:** 7
**Confidence:** 4

**Review:**

Overview: This paper proposes an outlier detection scheme based on contrastive self-supervised training for representation learning and cluster-conditioned detection using Mahalanobis distance. It shows 0.8% improvement in AUROC over the previous state-of-the-art by Winkens when trained using labeled in-distribution data. When training data is unlabeled, it performs significantly better than existing OOD detection algorithms based on AE, VAE, PixelCNN++, Deep-SVDD, and Rotation-loss and it performs only slightly worse than the labeled case. This is not surprising since I don't believe label information is crucial for OOD detection anyway, but it's good to confirm it. The authors show additional gain is possible when some OOD images are available during training, i.e., one or five images per class. This paper is very well written and I think the novelty and contributions are significant enough to merit its acceptance.

Comments: It's interesting to see the availability of only 1~5 samples per class for OOD data can result in some noticeable gains. I am not sure if covariance matrices have some scaling factors in (3), but how about using different weights in (3) for in-distribution and OOD data to take care of differences between sample sizes of in-distribution and OOD data? Would it make sense to have a higher weight for OOD data since it has fewer samples than in-distribution data?

There's a big performance gap between existing unlabeled schemes and the proposed scheme. Which one contributes most to the difference? NT-Xent? Mahalanobis distance? Cluster-conditioned detection? Some ablation studies that remove one or more of these would be interesting.

Authors say "higher eigenvalues dominates Euclidean distance but are least helpful for outlier detection", but wouldn't too small eigenvalues be less helpful for outlier detection because they are more sensitive to noise? Can authors provide an explanation on why higher eigenvalues are least helpful for outlier detection? It's strange that very small eigenvalues give very high AUROC values in Fig. 2.

In (3), why there's no minimization over m as in (2)? Is single cluster assumed for both in-distribution and OOD data in (3)?

Reference (SimCLR) missing for NT-Xent in equation (1).

Please consider including and comparing with the following references.

[X1] Denouden, et al., Improving Reconstruction Autoencoder Out-of-distribution Detection with Mahalanobis Distance, https://arxiv.org/abs/1812.02765

[X2] Vernekar, et al., Out-of-distribution Detection in Classifiers via Generation, https://arxiv.org/abs/1910.04241

[X3] Goyal, et al., DROCC: Deep Robust One-Class Classification, https://arxiv.org/abs/2002.12718

---

> ### Author Response · Authors · 2020-11-17
> **Response to Reviewer-3**
>
> Thank you for your thoughtful and constructive feedback. We would also like to thank you for supporting our work. We address your comments and questions below.
>
> **It's interesting to see the availability of only 1~5 samples per class for OOD data can result in some noticeable gains. I am not sure if covariance matrices have some scaling factors in (3), but how about using different weights in (3) for in-distribution and OOD data to take care of differences between sample sizes of in-distribution and OOD data? Would it make sense to have a higher weight for OOD data since it has fewer samples than in-distribution data?**
>
> Since the Covariance matrix uses an expectation over samples, we believe that a different number of in-distribution, and OOD samples will not cause a scaling issue in Equation (3). However, irrespective of that, a scaling factor between the distance from in-distribution and OOD clusters can be helpful (depending on the datasets). To validate it, we modify the equation (3) by introducing the scaling factor $C$ i.e., the modified equation (4) is following $ s_x = (z_x - \mu_{in})^T \Sigma_{in}^{-1} (z_x - \mu_{in}) - C \times (z_x - \mu_{U})^T S_{U}^{-1} (z_x - \mu_{U})$. We ablate $C$ from 0 to 2. With CIFAR-10 as in-distribution and varying OOD dataset, we achieve the following AUROC values.
>
> |           	| 0    	| 0.25 	| 0.5      	| 0.75     	| 1.0 (default) 	| 1.25 	| 1.5  	| 2    	|
> |-----------	|------	|------	|----------	|----------	|---------------	|------	|------	|------	|
> | CIFAR-100 	| 90.6 	| 91.6 	| 92.4     	| **92.7** 	| 92.6          	| 92.6 	| 92.2 	| 89.3 	|
> | SVHN      	| 99.6 	| 99.8 	| **99.9** 	| 99.7     	| 99.7          	| 99.3 	| 99.3 	| 99.1 	|
> | Texture   	| 97.6 	| 99.2 	| **99.4** 	| 99.3     	| **99.4**      	| 97.8 	| 97.6 	| 97.3 	|
>
> We avoid introducing such a scaling factor, as it would need access to another set of outliers to calibrate the scaling factor for each OOD dataset.
>
>
> **There's a big performance gap between existing unlabeled schemes and the proposed scheme. Which one contributes most to the difference? NT-Xent? Mahalanobis distance? Cluster-conditioned detection? Some ablation studies that remove one or more of these would be interesting.**
> While we indeed benefit largely from improvements in self-supervised representation, our cluster-conditioned detection approach is also critical to achieving state-of-the-art results. We conduct the following ablation study to justify our claim. We are using CIFAR-10 as in-distribution and CIFAR-100 as the OOD dataset.
> 1. We first test our detector on AE and VAE representations. Here both our detector and others, such as kNN and LOF, only achieve an AUROC close to 50. Such detection performance is likely due to poor representations learned by these methods, as the test accuracy over AE/VAE representations is only ~55% on CIFAR10 dataset.
> 2. Next we ablate on representations learned by SimCLR, which achieve ~92% accuracy on CIFAR10 dataset. First, we compare our choice of Mahalanobis distance with other distance metrics. Next, we compare out detection methods with two classical (kNN and LOF) and previous state-of-the-art (CSI [1]) outlier detection methods. Under both groups, our approach achieves the best results.
>
>            Distance Metric in our method       Outlier detection method
>       Euclidean Manhattan  Cosine Mahalanobis       kNN  LOF  CSI  Ours
> AUROC    89.23   86.98  89.27  **90.63**       90.48  90.42   90.0  **90.63**

---

> > ### Author Response · Authors · 2020-11-17
> > **Continuation of response to Reviewer-3**
> >
> > **Authors say "higher eigenvalues dominates Euclidean distance but are least helpful for outlier detection", but wouldn't too small eigenvalues be less helpful for outlier detection because they are more sensitive to noise? Can authors provide an explanation on why higher eigenvalues are least helpful for outlier detection? It's strange that very small eigenvalues give very high AUROC values in Fig. 2.**
> > We observe that eigenvalues of features of training data often lie in the range of 10^-4 to 10^1. While the in-distribution data span all eigenvectors we observe that out-of-distribution data has only a small component across eigenvectors corresponding to smaller eigenvalues. This effect leads to better detection of out-of-distribution data across eigenvectors with smaller eigenvalues. Euclidean distance weights all directions equally and thus fails to take advantage of this effect. Mahalanobis distance scales the euclidean distance appropriately by dividing with eigenvalues, and thus takes advantage of better separability across eigenvectors of smaller eigenvalues.
> > **Sensitivity to noise.** Note that the AUROC along a random unit-norm vector is 56.5 in the setup of figure 2. This number is significantly higher than AUROC along dominant eigenvectors. So yes, the presence of noise will influence smaller eigenvalues and eigenvectors. However, under severe noise, even when these eigenvectors will correspond to random directions, they will remain highly effective (compared to dominant eigenvectors) in outlier detection.
> >
> > **Comparison with additional previous works**
> > We have added a comparison with thirteen previous anomaly detectors, including DROCC [1],  in Table 3. Our approach achieves better performance than previous detectors. We didn't compare with Vernekar et al. [2] and  Denouden et al. [3], as they had results only on two smaller scale datasets, i.e.,  MNIST and FMNIST. In contrast, our approach scales to much larger datasets, such as ImageNet.
> >
> >
> > 1. Denouden, et al., Improving Reconstruction Autoencoder Out-of-distribution Detection with Mahalanobis Distance, https://arxiv.org/abs/1812.02765
> > 2. Vernekar, et al., Out-of-distribution Detection in Classifiers via Generation, https://arxiv.org/abs/1910.04241
> > 3. Goyal, et al., DROCC: Deep Robust One-Class Classification, https://arxiv.org/abs/2002.12718

---

### Official Review · AnonReviewer4 · 2020-10-28
**Borderline - could use additional analysis and ablations to separate improvements based on their method from improvements due to the SimCLR self-supervised representation**

**Rating:** 6
**Confidence:** 4

**Review:**

The authors tackle out-of-distribution detection without requiring additional detailed labels by leveraging the recent advancements in self-supervised methods combined with Mahalanobis distance metrics in feature space. They find that their approach matches or slightly outperforms supervised baselines, and widely outperforms other unsupervised approaches. They also consider a few-shot version of the task and extend their method to take advantage of supervision when available.

Pros:
Performance on outliers is a significant challenge for parametric models, so if outliers can be detected accurately this may help address potential failure modes in real-world systems. There is clear motivation for the problem.
The authors show clear gains over existing baselines.

Cons:
Their method is a pretty-straightforward combination of two established methods, contrastive self-supervised representation learning and the Mahalanobis distance as a metric of distance from a point to a distribution.
Their method seems to rely heavily on the efficacy of contrastive self-supervised learning. It would be good to see an ablations showing the performance of the Mahalanobis distance-based approach with the same embedding features as some of the poorly-performing unsupervised baseline methods to understand how much of the gains are due to the more powerful representation method. For example, could the authors add Mahalanobis distance on top of the learned representations from the autoencoder or VAE models? I know that the authors might have to get creative here, but as it stands it is not clear to me how much of these performance gains in the unsupervised setting are due to the author’s contribution, as opposed to just due to SimCLR.


Nits:

 Naming their method SSD seems unnecessarily confusing due to the popularity of the Single-shot multibox detector (SSD) model. I would recommend rethinking the acronym for clarity.

I would prefer to see the related work discussed before the authors proposed approach, so that their method can be better placed into context with the existing literature by the reader.

“Scaling with eigenvalues remove the bias” -> Scaling with eigenvalues removes the bias

---

> ### Author Response · Authors · 2020-11-17
> **Response to Reviewer-4**
>
> Thank you for your thoughtful and constructive feedback. We address your comments and questions below.
>
> **Their method seems to rely heavily on the efficacy of contrastive self-supervised learning. It would be good to see an ablations showing the performance of the Mahalanobis distance-based approach with the same embedding features as some of the poorly-performing unsupervised baseline methods to understand how much of the gains are due to the more powerful representation method.**
> While we indeed benefit largely from improvements in self-supervised representation, our cluster-conditioned detection approach is also critical to achieving state-of-the-art results. As suggested by the reviewer, we conduct the following set of experiments to justify it. We are using CIFAR-10 as in-distribution and CIFAR-100 as the OOD dataset.
> 1. We first test our detector on AE and VAE representations. Here both our detector and others, such as kNN and LOF, only achieve an AUROC close to 50. Such detection performance is likely due to poor representations learned by these methods, as the test accuracy over AE/VAE representations is only ~55% on the CIFAR-10 dataset.
> 2. Next we ablate on representations learned by SimCLR, which achieve ~92% accuracy on the CIFAR-10 dataset. First, we compare our choice of Mahalanobis distance with other distance metrics. Next, we compare our detection method with two classical (kNN and LOF) and previous state-of-the-art (CSI [1]) outlier detection methods. Under both groups, our approach achieves best results.
>
>            Distance Metric in our method       Outlier detection method
>       Euclidean Manhattan  Cosine Mahalanobis       kNN  LOF  CSI  Ours
> AUROC    89.23   86.98  89.27  **90.63**       90.48  90.42   90.0  **90.63**
>
>
> **Naming their method SSD seems unnecessarily confusing due to the popularity of the Single-shot multibox detector (SSD) model. I would recommend rethinking the acronym for clarity.**
> Thanks for pointing this out. We will certainly rethink a more suitable acronym for this work.
>
>
> **I would prefer to see the related work discussed before the authors proposed approach, so that their method can be better placed into context with the existing literature by the reader.**
> As suggested by the reviewer, we have moved the related work section earlier in the paper.
>
>
> **“Scaling with eigenvalues remove the bias” -> Scaling with eigenvalues removes the bias**
> We have corrected this in our updated paper.
>
> 1. Tack, J., Mo, S., Jeong, J., & Shin, J. (2020). Csi: Novelty detection via contrastive learning on distributionally shifted instances. Advances in Neural Information Processing Systems, 33.

---

### Official Review · AnonReviewer2 · 2020-10-28

**Rating:** 6
**Confidence:** 5

**Review:**

[Summary]
The paper addresses the OOD problem without learning from class labels. It proposes to learn the feature representation with unsupervised learning, then applies Mahalanobis distance to measure how far a test data is away from the in-distribution data. The proposed framework also considers the cases of when class labels or OOD data are available. The experiments on image datasets (mainly cifar-10/100) demonstrate its advantage over other unsupervised OOD methods.

[Reasons for score]
Bullet 1 in the cons is a significant concern. However, the method and analysis provide interesting insights. Overall, the pros and cons are equally substantial.

[Strengths]
1. The proposed framework is sounded and can well handle a variety of cases when class labels or OOD data are available.
2. The way it utilizes a small amount of OOD data is novel (eq.3).
3. The experiments and analysis are sufficient to support its major claim while providing additional insights (ex: Figures 2 and 4, Table 4).

[Weaknesses]
1. The eq.3 assumes OOD data are from a cluster that has the same mean and variance. This assumption is not valid in general cases. The experiments have a setup that favors this assumption; therefore, bias exists. The paper should explicitly point out this as a substantial limitation and provide some experiments (ex: use SVHN+CIFAR10 for the labeled OOD data) to elaborate it.
2. The evaluations are mainly on toy datasets. Although appendix table 6 uses ImageNet, it will be interesting if more realistic OOD datasets are included.
3. The use of the word “unlabeled” sometimes is confusing. There are two types of labels: class label and in/out-of-distribution label. The paper should clearly say its method is doing OOD without class labels. Please consider polishing the use of these terms in sections 1 and 2.

======================
POST REBUTTAL

The updated results make sense. The limitation/assumption mentioned in weaknesses 1 must be sufficiently disclosed in section 3.2.

---

> ### Author Response · Authors · 2020-11-17
> **Response to Reviewer-2**
>
> Thank you for your thoughtful and constructive feedback. We address your comments and questions below.
>
> **The eq.3 assumes OOD data are from a cluster that has the same mean and variance. This assumption is not valid in general cases. The experiments have a setup that favors this assumption; therefore, bias exists. The paper should explicitly point out this as a substantial limitation and provide some experiments (ex: use SVHN+CIFAR10 for the labeled OOD data) to elaborate it.**
>
> i) Yes, in the few shot OOD detection we assume access to outliers from the **targeted** OOD dataset, i.e., OOD dataset used at inference. We earlier state this assumption in Section 3.2. Based on the reviewer's concern we have added further clarification in Section 3.2.
>
> ii) Note that our baseline detection framework **doesn't require** access to outliers from the targeted OOD datasets (Section 3.1). But, if just a few samples are available, it can take advantage of it to boost performance.
>
> iii) We use few-shot OOD detection (Equation 3, 4) only when outliers from the targeted OOD dataset are available. We make this assumption as even a small number of outliers can allow us to approximate some statistics of the distribution of the OOD dataset. However, if outliers from a different dataset are used a training it will likely hurt performance as the statistics captured from training time outliers will differ from the distribution of the OOD dataset at test time.
>
> iv) As suggested by the reviewer, we conduct further experiments with the setup when available outliers at training time aren't from the OOD dataset used at inference. Consider the setup of CIFAR10 as in-distribution and CIFAR-100 as the OOD dataset. With the baseline SSD detector, we achieve an AUROC of 90.6. Now with five samples available from each class of CIFAR-100, it improves to 92.6. However, when the same amount of samples are available from the TinyImages dataset, which has different classes compared to CIFAR-100, it only achieves an AUROC of 89.9. Similarly, the approach only achieves an AUROC of 79.0 when we use textured images (from the DTD dataset) as outliers in training and test with outliers from CIFAR100. In summary, we assume that available outliers are from the targeted dataset itself as using outliers from a non-targeted dataset leads to performance degradation. In absence of such outliers (Section 3.1), our approach is still highly effective and achieves state-of-the-art performance (Section 4.2).
>
>
>
> **The evaluations are mainly on toy datasets. Although appendix table 6 uses ImageNet, it will be interesting if more realistic OOD datasets are included.**
> We are unsure of whether the reviewer is asking to use more OOD datasets for each in-distribution dataset or expand the list of in-distribution datasets, such as ImageNet, itself. Therefore we did both.
>
> i) First we added two more OOD datasets, namely LSUN and Places 365 for each in-distribution dataset. For ImageNet, we now also report results with ImageNet-O [1] as the OOD dataset.  We have updated Table-2 in the paper to reflect these results.
> ii) Second, we also experiment with tiny-ImageNet as the in-distribution dataset. Here SSD achieves an AUROC of 80.4, 93.8, and 91.1 for Texture, SVHN, and Blobs images as OOD datasets. With five-shot OOD detection, AUROC further improves to 84.9, 99.9, and 99.9, respectively. We will continue to experiment with more in-distribution datasets and update Table-2 accordingly in the camera-ready version of our paper.
>
>
> **The use of the word “unlabeled” sometimes is confusing. There are two types of labels: class label and in/out-of-distribution label. The paper should clearly say its method is doing OOD without class labels. Please consider polishing the use of these terms in sections 1 and 2.**
> We thank the reviewer for catching this issue. Indeed a standalone use of the word "unlabeled" can be ambiguous in this framework. We have clarified it early on in Section 1 to avoid this confusion.
>
> 1. Hendrycks, D., Zhao, K., Basart, S., Steinhardt, J., & Song, D. (2019). Natural adversarial examples. arXiv preprint arXiv:1907.07174.

---

### Official Review · AnonReviewer1 · 2020-10-29
**Limited novelty; major issues in experiment settings; important references are missing**

**Rating:** 6
**Confidence:** 5

**Review:**

This work investigates a classic unsupervised outlier detection problem, in which we do not have any label information and need to learn a detection model from those unlabeled data to identify any inconsistent data points as outliers. The key approach here is to apply existing self-supervised contrastive feature learning methods to extract feature representations and then apply a cluster-based method to calculate outlier scores. It also presents two ways to leverage labeled outlier data if available, including an improved mahalanobis distance method and the application of supervised contrastive learning methods proposed recently. The methods, including unsupervised and semi-supervised methods that use a few labeled outlier data, are evaluated using four datasets. As I read through the paper, I find the following major issues.

1. the question this work intends to answer, "Can we design an effective outlier detector with access to only unlabeled data from training distribution?", is a classic and well-studied problem in the anomaly/outlier detection community. There have been many studies over this problem. We cannot simply ignore those previous work  and states it as a new problem. see resources like Chandola, V., Banerjee, A., & Kumar, V. (2009). Anomaly detection: A survey. ACM computing surveys (CSUR), 41(3), 1-58. or Aggarwal, C. C. (2015). Outlier analysis. Springer, Cham. for numerous previous work on using shallow methods to address this problem.

2. There are a number of studies on self-supervised outlier detection approaches as well as what is called few-shot outlier detection approaches, but I cannot find any discussion of those work and the empirical comparison to these methods. The authors may refer to some recent survey papers, such as "Pang, G., Shen, C., Cao, L., & Hengel, A. V. D. (2020). Deep Learning for Anomaly Detection: A Review. arXiv preprint arXiv:2007.02500.", to find some of these studies. Some closely related methods are: self-supervised methods such as GT and E3Outlier that learns feature representations using a pre-text task in a self-supervised way; unsupervised outlier detection methods such as RDA, REPEN, ALOCC, OCGAN, etc.; methods that use a few labeled outlier data such as Deep SAD, DevNet, REPEN, etc. Please see table 1 in that survey paper for more details. The authors are suggested to discuss and differentiate their method from these existing studies, and to compare empirical comparisons to these closely related methods.

3. The claiming of "a parameter-free detector" is misleading and incorrect. Similar to other methods, the presented methods still have a huge number of hyper-parameters. They may be able to work without parameter tuning on each dataset, but this is not parameter-free. Also, many existing methods can also work in this way.

4. The way that the presented outlier detection methods utilizes the labeled outlier data may less effective than previous work, because the cluster-based anomaly scoring here is separated from the representation learning. More advanced approaches (see some of the methods mentioned above) can unify the anomaly scoring and the representation learning together and the labeled outlier data is used to optimized the entire anomaly detection pipeline, rather than the representation learning stage only.

5. The experiment settings are not properly designed to justify the paper's arguments. First, closely related deep unsupervised and semi-supervised methods are missing in the comparison in tables 1 and 3. Second, I think it is important to include some popular baselines here. For example, how is the performance of using traditional outlier detectors such as iforest, lof, and knn distance on feature representations extracted with a pre-trained resnet-50? How many benefits does the computationally extensive contrastive representation learning gain compared to those simple solutions?  Third, why is the cluster-based outlier scoring method used? can we use other traditional outlier detectors such as iforest, lof, and knn distance?

---

> ### Author Response · Authors · 2020-11-17
> **Response to Reviewer-1**
>
> Thank you for your thoughtful and constructive feedback. We address your comments and questions below.
>
> `The distinction between earlier works on out-of-distribution detection (OOD) and anomaly detection`: Before addressing the reviewer's concerns we would like to highlight the two different research directions in outlier detection.
> **Setup-A: Out-of-distribution (OOD) detection.** This recent line of work [1,2,3,4] considers a multi-class dataset as in-distribution and samples outliers from *another* multi-class dataset. It is often referred to as out-of-distribution (OOD) detection in the related literature.
> **Setup-B: Anomaly detection.** This direction is more focused on single class anomaly detection, where we consider one class of a dataset as in-distribution and the rest of the classes as a source of outliers [5,6,7].
> We focus on the former scenario, i.e., Setup-A, as detectors successful for anomaly detection (such as VAE, AE, PixelCNN) often perform poorly in the multi-class OOD detection setup (Table 1).
>
> We thank the reviewer for raising the issue of missing connection with work on anomaly detection. While our paper focuses on the former scenario (Setup-A), following the reviewer's feedback we now provide detailed experimental results (in response to the second comment) and show that our detector is also highly successful in this single-class anomaly detection setup. We hope that our results addresses the reviewer's concerns and hopefully close the gap between these two directions.
>
> **"Can we design an effective outlier detector with access to only unlabeled data from training distribution?", is a classic and well-studied problem in the anomaly/outlier detection community. There have been many studies over this problem. We cannot simply ignore those previous works and states it as a new problem.**
> Yes, we agree that unsupervised outlier detection is a well-established research problem. Our claim is certainly not that we are the first to solve this problem (Section 1, 3.1 and Table 1).  However, our comparison and references were based only on previous work on *OOD detection*. We have updated our discussion of related work to include broader anomaly detection literature.
>
> **There are a number of studies on self-supervised outlier detection approaches as well as what is called few-shot outlier detection approaches ... comparison to these methods.**
> We have now added a detailed discussion and comparison with these classes of detectors in Table 3 of the updated paper. We also provide a copy of the table below. Note that even without few-shot detection, our baseline detector (SSD) outperforms the previous work by a large margin.
>
> Comparison with CSI: A concurrent work from Tack et al. [8] proposes a detector for both OOD and anomaly detection. We outperform CSI in OOD detection, both with the absence and presence of training data labels (Table 1, 4), and achieve competitive performance in anomaly detection (Table 3). Note that unlike CSI, which utilizes multiple test time data augmentations, our approach works with unmodified test data.
>
>
> |    Method   | Airplane | Automobile | Bird |  Cat | Deer |  Dog | Frog | Horse | Ship | Truck | Average |
> |:-----------:|:--------:|:----------:|:----:|:----:|:----:|:----:|:----:|:-----:|:----:|:-----:|:-------:|
> | Random-Init |   77.4   |    44.1    | 62.4 | 44.1 | 62.1 | 49.6 | 59.8 |  48.0 | 73.8 |  53.7 |   57.5  |
> |     VAE     |   70.0   |    38.6    | 67.9 | 53.5 | 74.8 | 52.3 | 68.7 |  49.3 | 69.6 |  38.6 |   58.3  |
> |    OCSVM    |   63.00  |    44.0    | 64.9 | 48.7 | 73.5 | 50.0 | 72.5 |  53.3 | 64.9 |  50.8 |   58.5  |
> |    AnoGAN   |   67.1   |    54.7    | 52.9 | 54.5 | 65.1 | 60.3 | 58.5 |  62.5 | 75.8 |  66.5 |   61.8  |
> |   PixelCNN  |   53.1   |    99.5    | 47.6 | 51.7 | 73.9 | 54.2 | 59.2 |  78.9 | 34.0 |  66.2 |   61.8  |
> |    DSVDD    |   61.7   |    65.9    | 50.8 | 59.1 | 60.9 | 65.7 | 67.7 |  67.3 | 75.9 |  73.1 |   64.8  |
> OCGAN  | 75.7 	 |	53.1 	|	64.0 	|62.0 	|72.3 |	62.0 	|72.3 |	57.5 		|82.0 	|55.4   |  	65.6
> |     RCAE    |   72.0   |    63.1    | 71.7 | 60.6 | 72.8 | 64.0 | 64.9 |  63.6 | 74.7 |  74.5 |   68.2  |
> |    DROCC    |   81.7   |    76.7    | 66.7 | 67.1 | 73.6 | 74.4 | 74.4 |  71.4 | 80.0 |  76.2 |   74.2  |
> |   Deep-SAD  |    --    |     --     |  --  |  --  |  --  |  --  |  --  |   --  |  --  |   --  |   77.9  |
> |  E3Outlier  |   79.4   |    95.3    | 75.4 | 73.9 | 84.1 | 87.9 | 85.0 |  93.4 | 92.3 |  89.7 |   85.6  |
> |      GT     |   74.7   |    95.7    | 78.1 | 72.4 | 87.8 | 87.8 | 83.4 |  95.5 | 93.3 |  91.3 |   86.0  |
> |    InvAE    |   78.5   |    89.8    | **86.1** | 77.4 | **90.5** | 84.5 | 89.2 |  92.9 | 92.0 |  85.5 |   86.6  |
> |     GOAD    |   77.2   |    96.7    | 83.3 | 77.7 | 87.8 | 87.8 | 90.0 |  **96.1** | **93.8** |  92.0 |   88.2  |
> |     Ours    |   **82.7**   |    **98.5**    | 84.2 | **84.5** | 84.8 | **90.9** | **91.7** |  95.2 | 92.9 |  **94.4** |   **90.0**  |

---

> > ### Author Response · Authors · 2020-11-17
> > **Continuation of response to Reviewer-1**
> >
> > **The claiming of "a parameter-free detector" is misleading and incorrect ...**
> > We either explicitly avoid the use of additional tuning-parameters (such as when combining SSL and Supervised loss in SSD+) or refrain from tuning the existing set of parameters for each OOD dataset. Following the reviewer's concern, we have updated the text to further clarify this point.
> >
> > **The way that the presented outlier detection methods utilize the labeled outlier data may less effective than previous work, because the cluster-based anomaly scoring here is separated from the representation learning ...**
> > Note that we assume access to only a few outliers. For example in one-shot detection, with CIFAR-10 as the OOD dataset, we assume access to a total of 10 outliers (one from each class). We believe that training millions of network parameters from such a small number of outliers risks overfitting, likely degrading the performance.  Note that our even baseline detector, which doesn't use any outlier information, itself outperforms previous work in this direction (such as Deep-SAD) by a large margin (Table 3).
> >
> >
> > **The experiment settings are not properly designed to justify the paper's arguments. First, closely related deep unsupervised and semi-supervised methods are missing in the comparison in tables 1 and 3.**
> > We again urge the reviewer to note the distinction between the experimental setup of earlier works on OOD detection and anomaly detection. While the former consider outliers from different datasets, the latter works samples anomalies from a different class of the same dataset. As we reported in Table 1, classical anomaly detection methods which work well on modeling single class, such as PixelCNN and Deep-SVDD, often fail in the setup of OOD detection. However, the state of the art in anomaly detection (CSI [8]) also reports competitive performance in outlier detection. We have added a comparison with this method in both Tables 1 and 3. In both places, we outperform CSI in most cases.
> >
> >
> > **Second, I think it is important to include some popular baselines here. For example, how is the performance of using traditional outlier detectors such as iforest, lof, and knn distance on feature representations extracted with a pre-trained ResNet-50?**
> > We find that features from pre-trained networks are good but competitive enough. Consider CIFAR-10 as distribution with CIFAR-100 and SVHN as OOD datasets. Here features extracted from a ResNet-50 network pre-trained on ImageNet achieves AUROC of 81.7 and 97.8, respectively (we report the best numbers achieved across iforest, lof, and knn). In contrast, our approach achieves AUROC of 90.6 (+8.9) and 99.6 (+1.8), respectively.
> >
> >
> > **Third, why is the cluster-based outlier scoring method used? can we use other traditional outlier detectors such as iforest, lof, and knn distance?**
> > In comparison to other detection methods, we find that our Mahalanobis distance-based cluster-conditional framework achieved the best results. Here we fix the representations, which are learned with NT-Xent loss, and vary the choice of the detection method.
> >
> >
> > | Method   	|   	| kNN (Euclidean) 	| kNN (Mahalanobis) 	|   	| LOF (Euclidean) 	| LOF (Mahalanobis) 	|   	| Ours (Euclidean) 	| Ours(Mahalanobis) 	|
> > |----------	|---	|-----------------	|-------------------	|---	|-----------------	|-------------------	|---	|------------------	|-------------------	|
> > | ResNet18 	|   	|      89.07      	|       89.15       	|   	|      81.97      	|       88.46       	|   	|       87.28      	| 89.50             	|
> > | ResNet50 	|   	|      89.95      	|       90.47       	|   	|      83.65      	|       90.42       	|   	|       89.32      	| **90.63**            	|
> >
> > 1. Liang, S. et al. (2018, February). Enhancing The Reliability of Out-of-distribution Image Detection in Neural Networks. In International Conference on Learning Representations.
> > 2. Winkens, J. et al. (2020). Contrastive training for improved out-of-distribution detection. arXiv preprint arXiv:2007.05566.
> > 3. Hendrycks, D., & Gimpel, K. (2017). A baseline for detecting misclassified and out-of-distribution examples in neural networks.  In International Conference on Learning Representations.
> > 4. Lee, K. et al. (2018, February). Training Confidence-calibrated Classifiers for Detecting Out-of-Distribution Samples. In International Conference on Learning Representations.
> > 5. Perera, P. et al. (2019). Ocgan: One-class novelty detection using gans with constrained latent representations. In Proceedings of the IEEE Conference on Computer Vision and Pattern Recognition (pp. 2898-2906).
> > 6. Chalapathy, R. et al.(2018). Anomaly detection using one-class neural networks. arXiv preprint arXiv:1802.06360.
> > 7. Golan, I., & El-Yaniv, R. (2018). Deep anomaly detection using geometric transformations. In NeurIPS.
> > 8. Tack, J. et al. (2020). Csi: Novelty detection via contrastive learning on distributionally shifted instances. NeurIPS

---

### Decision · Program_Chairs · 2021-01-07
**Final Decision**

**Decision:**

Accept (Poster)

**Comment:**

There is some positive consensus on this paper, which improved somewhat after
the very detailed rebuttal comments by the authors. The use of limited amounts of OOD data is interesting and novel. There were some experimental design problems, but these were well-addressed in rebuttal.

A reviewer points out that
anomaly/outlier detection does not explicitly assume that there is only one
class within the normal class (or in-distribution data). The one-class
assumption is mainly made in some popular anomaly detection methods, such as
one-class classification-based approaches for anomaly detection. The authors
should take this into careful consideration when preparing a final version of
this work.